# *Who am I?* Narratives as a window to transformative moments in critical care

**Briseida Mema**[1,2¤]*, **Andrew Helmers**[1,2], **Cory Anderson**[1], **Kyung–Seo (Kay) Min**[3], **Laura E. Navne**[4,5]

**1** Department of Critical Care Medicine, Hospital for Sick Children, Toronto, Canada, **2** Department of Pediatrics, Faculty of Medicine, University of Toronto, Toronto, Canada, **3** History of Art Department, Johns Hopkins University, Baltimore, MD, United States of America, **4** The Danish Center for Social Science Research, VIVE, Copenhagen, Denmark, **5** The Department of Public Health, University of Copenhagen, Copenhagen, Denmark

¤ Current address: Department of Critical Care Medicine, The Hospital for Sick Children, Toronto, ON, Canada

* briseida.mema@sickkids.ca

**Data Availability Statement:** All relevant data are within the paper and its Supporting Information files.

**Funding:** BM received the Medical Humanities Education Grant, 2020 from the University of

## Abstract

Critical care clinicians practice a liminal medicine at the border between life and death, witnessing suffering and tragedy which cannot fail to impact the clinicians themselves. Clinicians' professional identity is predicated upon their iterative efforts to articulate and contextualize these experiences, while a failure to do so may lead to burnout. This journey of self-discovery is illuminated by clinician narratives which capture key moments in building their professional identity. We analyzed a collection of narratives by critical care clinicians to determine which experiences most profoundly impacted their professional identity formation. After surveying 30 critical care journals, we identified one journal that published 84 clinician narratives since 2013; these constituted our data source. A clinician educator, an art historian, and an anthropologist analyzed these pieces using a narrative analysis technique identifying major themes and subthemes. Once the research team agreed on a thematic structure, a clinician-ethicist and a trainee read all the pieces for analytic validation. The main theme that emerged across all these pieces was the experience of existing at the heart of the dynamic tension between life and death. We identified three further sub-themes: the experience of bridging the existential divide between dissimilar worlds and contexts, fulfilling divergent roles, and the concurrent experience of feeling dissonant emotions. Our study constitutes a novel exploration of transformative clinical experiences within Critical Care, introducing a methodology that equips medical educators in Critical Care and beyond to better understand and support clinicians in their professional identity formation. As clinician burnout soars amidst increasing stressors on our healthcare systems, a healthy professional identity formation is an invaluable asset for personal growth and moral resilience. Our study paves the way for post-graduate and continuing education interventions that foster mindful personal growth within the medical subspecialties.

Toronto.The funders had no role in study design, data collection and analysis, decision to publish, or preparation of the manuscript.

## Introduction

". . . wanderers in the wilderness of disease hoping to find a path to peace . . ." (Pinsky, 2013)

Science and Art complement each other in our efforts to better understand the human condition [1]. The effort to examine, reflect upon, and articulate the human experiences of health, sickness, and death, constitutes the field of *medical humanities* and exemplifies the convergence of Science and Art intrinsic to every clinical encounter [2]. While the entire field of medical humanities shares this goal of applied humanism, it is nevertheless a diverse field, comprised of different perspectives and methodologies which together bring new insights to medical education [3, 4]. In particular, narrative medicine has emerged as a way for clinicians to engage in self-reflection and to frame their medical knowledge within a mindful appreciation of the patient-as-person [5–7]. Through reflective writing, clinicians examine the personal journeys undertaken by their patients and by themselves, through a process of inquiry, discovery, and analysis [8, 9]. As they navigate challenging situations which transcend textbook knowledge, clinicians learn (and teach, through their writing) how to forge meaning amidst a landscape that is conflicted and emotional more than it is analytic [10, 11]. Writing these narratives becomes a way of reconciling the clinicians' personal journeys with a profession that can be complex and ambiguous; a way of discovering–and illuminating for others–how to forge a professional identity through reflection on how that profession affects patients and clinicians alike [12–14].

The question of what makes someone into the clinician they become never ceases to be relevant in medical education, and perhaps now more than ever before there is an urgency to ask and answer that question. Today's clinical milieu has become better attuned (albeit imperfectly) to the importance of wellness, moral distress, and burnout among clinicians; and has become (even more imperfectly) cognizant of the determinants of patient care that defy mere medical rote. In this context, medical education must concern itself more consciously with the longitudinal professional identity formation of clinicians at all stages from early training to transitioning out of practice. Narrative writing provides a unique insight (a shared learning) into the transformative experiences which shape clinicians' professional identity [15–18]. There is ample historical precedent for clinicians expressing themselves in fiction and in essays published in the lay and scientific press [19–21]. Today's medical journals are publishing an increasing number of medical narratives, forming a rich syllabus that reflects, shapes and makes visible the iterative process of clinicians' professional identity. Shared amongst a broader professional community, these narratives provide an opportunity for reflection among individuals with similar experiences; medical educators are thus provided with an opportunity to equip clinicians with a wealth of reflective experiences, to avoid the unexamined life and establish instead a collective self-awareness together with a more holistic understanding of their profession.

We were interested in the professional identity formation of critical care clinicians for the reasons that: (a) they are described to have one of the highest burnout rates [22] and (b) the context of their work typically brings patients and families face-to-face with mortality while blurring the boundaries between human and machine [23, 24]. Intensive care units (ICUs) form the setting for profound life-altering experiences; clinicians, patients and families are constantly challenged to muster the courage and generosity to tolerate and bear witness to unfair losses and tragedies. A comprehensive survey of narratives written by ICU clinicians may offer insight into the way clinicians make sense of their own professional journeys while

facing sick and dying patients and help identify core elements relevant to the process of professional identity formation within this environment. The goal of this study was to analyze these narratives, written by ICU clinicians, to answer: What are the experiences shared through narratives that impact ICU clinicians' identity formation? How is meaning sought in these experiences? What impact do these experiences have upon their personal and professional lives?

## Materials and methods

Narratives are published in medical journals, books, blogs, and different websites. Books are written by one author, and we were interested in the community of critical care clinicians. We didn't have a good method of identifying blogs and other websites and generally they are not peer reviewed, so we focused on narratives published in medical journals. In order to identify pieces published by ICU clinicians, we looked across 30 journals that publish content related to Intensive Care. We looked at each journal's "Instructions for authors" guidelines to find out if the journal published narrative pieces written by clinicians (S1 Appendix). We found one journal, that consistently published narrative pieces since 2013 in a section titled "From the Inside" [25]. We included all 84 reflective pieces published and these pieces constituted our data source.

Narratives, experiences written in story form, can provide insight into how people make sense of the world and how they communicate their insights to others, providing a "framework for our attempts to come to terms with the nature and conditions of our existence," as suggested by psychologists Jens Brockmeier and Rom Harre [26]. By conducting a qualitative analysis, using a narrative methodology and thematic data analysis, we sought primarily to understand how the reflective pieces helped the writer make sense of their key experiences [27, 28]. We treated the pieces as narrative pathways to the writer's identity formation [29]. The choice of a narrative methodology hinged on our preliminary readings of the reflective pieces. We noticed that the point of departure for the majority of these pieces is a transformative moment in a professional's critical care career, followed by reflections and examples of how such moments have come to shape not only their career paths but also their personal views of their role as clinician in the process of living and dying.

Three research team members: a clinician and educator (BM), a medical anthropologist (LN) and an art historian (KM) participated in the initial coding and data analysis. We recognised that our analyses involved choices made by us–choices about which pieces to highlight or de-emphasize. Using thematic data analysis, we searched across stories for central themes or patterns. Conducting first descriptive and then interpretative coding, we identified major themes and subthemes, and then discussed, revisited, and compared our findings. Findings were discussed and analyses were revised at team meetings until a stable thematic structure was agreed upon. The pieces were frequently re-read throughout the analysis to ensure that the analysis stayed as close as possible to the described experiences and contextual factors that the writers portrayed in their stories. Once the research team agreed upon the thematic structure a clinician and ethicist (AH) and an ICU trainee (CA) read the pieces for analytic validation, to see if they recognized the same themes as central to the writers' experiences. Member check in was not realistic, considering the pieces were written by more than 100 international clinicians. Our team presented our finding to a group of more than 40 Intensive Care clinicians (doctors, trainees, nurses and respiratory therapists) in our institution and the conclusion was that the finding resonated with them all.

## Results

The reflective pieces all begin with transformative moments, encounters which personally challenge the clinician in a way that leads them to discover or re-discover who they are as a

person within their profession. The predominant theme across the narratives was the experience of existing at the heart of a dynamic tension between life and death. We identified three further sub-themes: the role dualism; the effort to be the same person on the inside while straddling the divergent worlds of the hospital and the home, and the internal conflict between concurrent but dissonant emotions. These themes are explored below, with illustrative quotations from selected essays in separate tables; our analysis is provided in (S2 Appendix).

## Main theme

**Existing at the heart of a dynamic tension between life and death.**   The reflective pieces consistently demonstrate that becoming an ICU clinician usually begins with a strong drive to save lives followed by a gradual realization that their profession has an important role in caring for patients who are dying. The majority of the pieces grapple with the imminence of death.

Life is illuminated by death as a treasure; patients cling to life with "hope against hope", even when one is breathless, and all the odds are stacked against its continuation. Impending death brings a love for life sharply into focus and lends an urgency to a search for meaning; clinicians become inexorably entangled in the efforts of patients and families to make and preserve memories and to weed out regrets.

In their narratives, clinicians struggle with a "good death". For some, the ICU can loom as an "arena" where death is fought as a medical condition–contingent, changeable–rather than a tragic reality of the human condition. Clinicians at times are privy to the horror of a prolonged dying process that inflicts progressive "violence" against the person such that death looms as relief from medicine's machinations.

The narratives grapple with the existential crisis which is a daily reality in critical care: how should I live? How should I die? And the clinicians, facing their own finitude, write about the small but important role they play in a "good death" by respecting the failing body, offering compassion, and relieving suffering. Knowledge and technology cannot stop patients' tears from falling, but humanism can help wipe tears away.

Faced with death, life takes on added intensity and meaning. The critical care narratives in our study describe how clinicians' face death with and through their patients, forging a new lens through which they see and shape their own lives, professional and personal. Detailed quotes are provided in Table 1.

## Subtheme 1

**Role dualism.**   The tension between life and death in critical care is sometimes experienced as a mirror rather than a lens; every clinician experiences for themselves and for their loved ones, moments where they assume a patient's role. These moments challenge and refine professional identity as clinicians witness and support suffering and grief while counting themselves among those who suffer and grieve. Experiencing medicine as a patient or patient-partner brings an enhanced awareness of the limits of medicine; judgments, predictions, objectivity–all of these professional habits are examined anew.

Tasked with high-stakes decisions about individuals' destinies in charged situations, clinicians reflect on the limits of their abilities and ask, what is humanly possible? Is it a human task to be able to answer the patient's questions: "am I going to die?" or "am I dead?" upon their awakening from coma? Does it require superhuman abilities to make judgements about quality of life and to decide when death is the "better option" in the face of ambiguity? Occupying the liminal space between self-confidence and doubt, the analytical and the emotional, clinicians have the unique opportunity to bridge the gap between objective data and human emotion. These clinicians gather the courage to recognize the superior role of chance and

**Table 1. Quotation for the main theme.**

| Existing at the heart of a dynamic tension between life and death |
|---|
| Life is something Wild, <br> Touching your ears, <br> While your mind tries to puzzle out the honest truth. (Ranzani, 2013) |
| Hoping against hope in the sea of despair where the hope only floats. <br> Breathing yet breathless, living yet lifeless. (Kumar, 2014) |
| Why open your chest to fix what we no longer could? <br> Why pump on your heart when it no longer would? <br> I wanted to contest death, and I believed I should. (Gusmao-Flores, 2020) |
| Every floor is one of several significant illness episodes. On his way down passing each floor, doctors got used to reassuring him saying: "So far so good". (Elia, 2018) |
| Modern medicine allows unimagined possibilities to save one's life and keep death at bay. (Bein, 2018) |
| But who was my patient? I pause here and acknowledge it was the bleeding, dying boy that I continued to maintain on multiple life-sustaining modalities. (Neville, 2015) |
| The question may come up ''why is it that it is so hard to die now?'' Is it that we have to exhaust all technologies before this is OK? (Bein, 2015) |
| That ''someday'' is today. I am here with you. I won't leave you. (Benbenishty, 2014) |
| Caressed by the icy hands, freed from the burning breaths. (Kumar, 2014) |
| But my humble gratefulness will be for that one <br> Who was warm and human (Galvez, 2013) |
| Let me find death Far, far away Into the heart of the mountains. (Hernandez, 2013) |
| Suddenly, the sky was broken by the red horn of the moon emerging from a cloudy horizon in the middle of my solitude. A crescent moon announcing what poets and mystics have claimed for centuries: life goes on no matter how many tragedies you have to endure. (Gorrea, 2013) |
| Every night can be our last. Make sure that the final goodbye will be a sweet sorrow. I learned this valuable lesson that night, 25 years ago. (Kompanje, 2019) |
| I had experienced a crucial life event, a sort of epiphany. I had encountered the fascinating alchemy between physiology and humanism. (Hernandez, 2013) |
| And yet, sometimes, often times, it is not practical medical acumen that serves my patients well, but rather a pure and simple compassion . . . wanderers in the wilderness of disease hoping to find a path to peace that if it could not mean recovery at least would not mean pain and distress. (Pinsky, 2013) |

context and take responsibility for the margin of difference their choices can make. Though some decisions may seem too difficult or inhumane it is an uncomfortable reality that these decisions can only be made by humans.

As patients themselves, as caregivers for a sick loved one, and as a consequence of empathy, clinicians find their professional habits tempered by personal humility. These dual experiences–to diagnose and to seek a diagnosis, to care and be cared for, to grieve with and to grieve for, to trust and to seek trust–become an opportunity for clinicians to affirm their profession as a vocation, strengthening their ability and capacity to "suffer with" that is intrinsic to compassion. Their role dualism thus becomes a way to help their patients navigate the dying process. Detailed quotes are provided in Table 2.

## Subtheme 2

**Bridging two worlds.** The Intensive Care Unit is a surreal place, its sights and sounds combining last century's anachronisms side-by-side with the stuff of science fiction. Clinicians work long hours in this place, and witness patients and families occupy an unnatural permanence in what is "meant" to be a transient waystation. For clinicians and patients alike, the ICU warps time and space: in one bed, time does not pass and becomes monotonous, the minutes resemble days with the unbearable disappointment that one is still in bed, still immobilized, physically or mentally; in the next bed, an emergency, where seconds of indecision or minutes of high-stakes work can seem like an eternity. The lights are always on, the patients and their families are always embroiled in the most important and even the final struggle of their life together. Clinicians' professional and personal lives can seem worlds apart; their

**Table 2. Quotation for the sub-theme 1: Role dualism.**

I froze as a sudden unwelcome surge of panic welled up inside of me and shattered my line of thought like a hammer hitting glass. She was my daughter. (Maclaren, 2013)

What is it like to grieve while working in the ICU? Is it possible to rise from the depths of your personal loss and help others? (Mema 2020)

It obliges us to deliver potentially devastating news about prognosis directly to an ill patient. In the process, we must face our own apprehensions about navigating difficult conversations and perhaps even contemplating our own mortality. (Isaac, 2017)

. . .frequently faced with judgements regarding the burdens and benefits of continuing artificial support—and life itself. (Koch, 2017)

Up to what point could the confidence we have in such prognosis—in clear conscience—legitimately modify the future of a piece of the mankind we have in our charge, represented by a single individual destiny? (Gobert, 2016)

Two patients showed me their capability to predict time of death without any knowledge of complicated scoring systems or many years of clinical experience. (Bakker, 2013)

This gives us the illusion of being able to fully control the biological aspects of our patients' existence. It almost appears that their lives depend on our interventions and decisions (Elia, 2018)

realized that choice was an illusion. (Maringer, 2018)

. crucial to find the balance between compulsive technological care or just listening to the patient's dreams. This was Don Dago for me, a man who lived and died like he wanted to. (Sánchez, 2014)

The shield that had allowed me to maintain a safe distance from the suffering around me was shattered, and I felt the sobering but no longer suffocating weight of responsibility that my profession conferred on me. (Gristina, 2013)

I cried. My "empathic filter" had failed completely. (Kompanje, 2019)

journey home each day can feel like they hold a privileged return ticket for the ferry across the Styx. On one shore: the monumental, the beatific, technology, extraordinary; on the other, the mundane, the banal, humane, personal.

Clinicians' narratives speak of their fraught efforts to assert personhood as a professional, to bring something personal into the impersonal tangle of machines at the bedside, to build bridges that preserve a single identity at home and in the hospital both for themselves and for their patients. Some invoke art and music, one brings a "swimming pool" for a bedside baptism, one helps bring an intubated child home for Christmas against the "rules"; together they speak to the importance of affirming personal identity and meaning amidst what can be a disorienting professional environment. This effort to *be* (and empower being) in a way that is not defined by the hospital environment becomes a defining part of the clinician-narrators' professional identity, a way of seeing patients and families that transcends the accoutrements of acute care medicine. Detailed quotes are provided in Table 3.

## Subtheme 3

**Emotional disparity.** Critical care clinicians experience and witness a myriad of emotions throughout each day–walking from room to room in the ICU might mean moving from a death vigil to a patient ready for discharge. Rooms become haunted with memories of patients' mortality, each death and each "triumph" leaving an inexorable mark. Such memories remain vivid after many years, and so the cacophony of daily emotions in the present are joined and shaped by emotions past: hope, despair, attachment, detachment, joy, and grief.

**Table 3. Quotation for the sub-theme 2: Bridging Two Worlds.**

There was what I could perceive of the real world around me, and then there was my dream world. (Fenn, 2014)

The sick, whom we shepherd through the shards of night, dream their jittery dreams, unconscious, scissor-step from light to nightmare and back again. (Butka, 2014)

"It's not her time," everyone said.
A full month since the hit-and-run,
We gave her a life, in an ICU bed. (Yan, 2019)

We are like in a submarine? He shows that there are no windows, no natural light, and no knowledge of night or day —no landmarks. (Nguyen, 2018)

Clinicians maintain hope while facing a new reality by recalibrating, looking at life and disease with a new perspective; but hope in critical care is a vulnerable thing, haunted by despair and regret. Joy emerges with a new intensity from situations that seemed hopeless but it cannot always shake the shadows of uncertainty and doubt.

In this tense context, clinicians feeling detachment may seem more protected from this vulnerability but still naturally get attached to their patients and "crave" to know the human in the ICU bed. Acknowledging the incomplete, fragile connection to their patients, they use music, touch and other forms of non-verbal communication that might reach their patients. They explore their fears and pains and try to relieve them and ascribe a personality to their patients. Sometimes these clinicians write biographies for their patients, depending on relatives to fill in the blanks or, if relatives are absent, becoming a kind of relative themselves. Clinicians find a way to deal with their own grief for the loss of a patient, often transforming it into personal growth, learning, and empathy.

Though the burden of conflicting emotions weighs heavily, many of the narratives invoke a concomitant sense of fulfillment, which stems from being able to support dying patients and their families, or accompanying them through their most vulnerable times, or helping a patient put on their shoes and witnessing them take their first walk after a long "pause." Fulfillment drives clinicians' resilience–their longevity within and dedication to this necessarily intense but fulfilling career. Detailed quotes are provided in Table 4.

## Discussion

We sought to examine longitudinal Professional identity formation (PIF) by analyzing narratives written by clinicians–trainees and senior learners alike–within Critical Care Medicine, a speciality that is responsible for some of the highest rates of burnout among its staff [30, 31]. It is an elemental and disquieting fact that we all will cease to exist at some point; however, for most it temporarily remains an abstraction facilitated by being caught up in the urgent matters of living. Forced by the circumstances of their profession, these clinicians meet the imposing reality of death and their professional demands living in the heart of the tension between life and death. To manage this tension requires living among opposing forces, events, roles, emotions, and contexts.

Our study adds to the literature in several ways. First, while PIF is described as a longitudinal and complex lifelong process, the majority of studies to date have focused on medical students [32–35]. The 84 reflective pieces in our analysis provide rich examples of key transformative experiences that impact the professional identity formation of this group of clinicians while they reflect on critical formative experiences to construct meaning and clarify

**Table 4. Quotation for the sub-theme 3: Emotional Disparity.**

. . .many times we want to turn around the inescapable course of the events. Somehow, someway, we get some hints that seem to endorse our innermost longings. (Castro, 2014)

It was so much easier then to hold your quiet hand
than it is now to keep your spirit standing tall.
But I am still here despite it all, always
sitting by your side. (Struwing, 2015)

I help her put on her socks and soft shoes, although she does not like the fold of the socks and asks me to fix them. As I am doing this, she is looking at me with approval and a knowing glance that she is ready to begin. (Polastri, 2015)

We stayed there to show respect. To recognize the girl who was there. To remember her life and laughs. Through silence, we tried to emphasize the brevity and importance of that moment. Because she deserved it as a human being. (Garcia-Salido, 2019)

We stopped rounds in front of room number seven as a signal of respect. We felt so sad and empty. So many weeks with us, and we barely knew this man—his dreams, hopes, and fears. (Hernandez, 2019)

values as individuals within their profession. Our analysis thus uncovers an important framework for medical educators to foster PIF within Critical Care, and also establishes a unique means by which other medical specialties can also add to their understanding of PIF.

Identity conflicts (demonstrated in roles, contexts, and emotions) can be a source of "tension" as personal meets professional and one strives for an idealized professional self. As evolving professionals confront new realities of medical practice that depart from their initial expectations, they must adapt and reconcile the "ideal" and lived versions of professional identity. Working at the center of these tensions can be disorienting and exhausting. There will always be the temptation to disconnect and retreat, to distance oneself from existential turmoil, and even to treat death as if it constitutes an adverse event. But the tension itself never disappears, and to ignore it is to risk the loss of perspective and professional integrity to which clinicians strive. These clinicians write about how they seek to live in this tension: to master it before it masters them; as a group, their pieces resemble "coming-of-age" stories with protagonists who undergo psychological and moral growth through their experience of the tempest that is clinical care. The narratives in our study promote an understanding of the transformation intrinsic to professional identity formation, holding up a mirror to each of us by articulating what it means to navigate those essential day-to-day tensions that challenge clinicians to question and renew their core values.

Concerns about burnout in critical care clinicians in particular and the medical community as a whole are growing rapidly, and many societies are calling for action [30, 31]. Research and policy development to address burnout are constrained by the inability to eliminate morally and emotionally challenging situations in the clinical realm; to truly mitigate burnout, medical practice must foster professional growth and moral resilience [36]. Professional identity formation has been associated with the development of an "inner compass" in trainees and practicing clinicians, together with resilience [37–41]. The goal of professional identity formation has been described as "to anchor students to foundational principles while helping them navigate the inevitable moral conflicts in medical practice" [42]. Role models, mentors and accumulation of clinical experiences are the most powerful factors that influence professional identity formation [16, 43, 44]. Transformative learning represents one process through which learners make sense of their key professional experiences; such learning utilizes reflections such as the narratives in our study, facilitated by a role model, mentor, or a peer group [11, 40, 45].

Our analysis engages with a complete, albeit relatively small, corpus of reflective narratives by intensive care professionals and highlights a shared exploration of the tension at the heart of the clinical work. These narratives serve as inspiration for ways in which intensive care trainees may be introduced to the field. The tensions described within these narratives reflect the common and disorienting experiences that impact professional identity formation, the raw material for medical educators to engage with as mentors and role models to guide reflection and to enrich each clinician's perspective [45]. Exposing these crucial moments provides opportunities to anticipate and support positive professional identity development. Moreover, such narrative writing may offer a vehicle for formal teaching; these pieces challenge each clinician to confront their clinical terrain, to chart a course consistent with who they are as a professional who sits across from their patient as a fellow traveller [46].

There are a few limitations to our study. First, our focus on a particular group might be construed as a limitation; however, the themes contained in our analysis will likely resonate beyond the field of Critical Care. Further, the authors of these pieces transcend the institutional and geographic barriers typical of most research, and thus reflect a further generalizability. We recognise that personal stories published in a journal are written for a particular audience, namely health professionals and colleagues who conceivably have a shared

understanding of the exigencies of the critical care profession. Moreover, we recognized that such stories will necessarily differ from those told to other audiences. At the same time, all narratives, including those gathered from selected journals, are incomplete and received differently over time, according to their context and by each reader. We acknowledge that this is a special group of individuals for many reasons: they are willing to share their dilemmas, challenges, and vulnerabilities with a readership of their peers; they are able to comprehend and analyze their feelings and express them beautifully; they describe key events that shape and transform the writer; and they analyze events and experiences as part of the larger picture of the clinician's emotional education beyond the personal parameters of the event. These individuals are also able to produce well-written narratives and use literary devices and a knowledge of literature to articulate their lives as art. In doing so, they shift the reader's perspectives and broaden their understanding. Finally, we realize that these pieces are selected and revised by external editors–nevertheless, it would be difficult to obtain a deliberate sampling of intensive care clinicians' writing, as many would not embrace this mode of expression for their inner reflections.

## Conclusions

In summary, our study provides insight into the longitudinal professional identity formation of clinicians working in intensive care. Utilizing narratives to make sense of complex human experiences, our analysis captures the way in which critical care clinicians build a mature understanding of their profession through transformative experiences. The themes identified herein can serve as a novel resource for medical educators, colleagues and leaders to build supportive, effective, and healthy interventions to foster emotional and moral resilience. It is our hope that the analysis we employed be emulated within other medical specialties in order to enhance their understanding and support professional identity formation, and that medical humanities in general be invoked to better understand the values essential for sustainable patient care.

## Supporting information

**S1 Appendix. List of journals that publish content related to Critical Care Medicine with corresponding website.**
(DOCX)

**S2 Appendix. Analyzes of narratives written by ICU clinicians.** (Narratives were published in the section "From the Inside", Intensive Care Medicine Journal).
(DOCX)

## Author Contributions

**Conceptualization:** Briseida Mema.

**Data curation:** Briseida Mema, Laura E. Navne.

**Formal analysis:** Briseida Mema, Andrew Helmers, Cory Anderson, Kyung–Seo (Kay) Min, Laura E. Navne.

**Funding acquisition:** Briseida Mema.

**Investigation:** Briseida Mema, Cory Anderson.

**Methodology:** Briseida Mema, Kyung–Seo (Kay) Min, Laura E. Navne.

**Project administration:** Briseida Mema.

**Resources:** Briseida Mema.

**Supervision:** Briseida Mema, Laura E. Navne.

**Validation:** Briseida Mema, Kyung–Seo (Kay) Min, Laura E. Navne.

**Visualization:** Briseida Mema.

**Writing – original draft:** Briseida Mema, Andrew Helmers.

**Writing – review & editing:** Briseida Mema, Andrew Helmers, Cory Anderson, Kyung–Seo (Kay) Min, Laura E. Navne.

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
