## [Decision Letter · Decision Letter 0]

9 Aug 2021

PONE-D-21-09314

Who Am I? Narratives as a Window to Transformative Moments in Critical Care

PLOS ONE

Dear Dr. Briseida Mema,

Thank you for submitting your manuscript to PLOS ONE. After careful consideration, we feel that it has merit but does not fully meet PLOS ONE’s publication criteria as it currently stands. Therefore, we invite you to submit a revised version of the manuscript that addresses the points raised during the review process.

Interesting study that explores a field of Narrative Medicine in Critical Care with a new approach.

However, to be publishable on PLOSONE it is necessary that the Authors answer the following points:

To respond accurately to the Reviewers’ requestsTo improve the methodology of the proposed studyTo better define the study designTo highlight how the results of the study can be valid in contexts other than the one where the study was carried out.Please use specifing reporting guideline and adhere to either COREQ, SRQR, or any other applicable EQUATOR guideline.Please Explain in the Discussione the acronym PIF.Please include the Limitations and Conclusions paragraphs in the paperTo improve the bibliography by adding more recent studies on the subject.

Thedecision is justified on PLOS ONE’s publication criteria.

We look forward to receiving your revised manuscript.

Kind regards,

Filomena Pietrantonio

Academic Editor

PLOS ONE

Additional Editor Comments (if provided):

Interesting study that explores a field of Narrative Medicine in Critical Care with a new approach.

However, to be publishable on PLOSONE it is necessary that the Authors answer the following points:

1. To respond accurately to the Reviewers’ requests

2. To improve the methodology of the proposed study

3. To better define the study design

4. To highlight how the results of the study can be valid in contexts other than the one where the study was carried out.

5. Please use specifing reporting guideline and adhere to either COREQ, SRQR, or any other applicable EQUATOR guideline.

6. Please Explain in the Discussione the acronym PIF.

7. Please include the Limitations and Conclusions paragraphs in the paper

8. To improve the bibliography by adding more recent studies on the subject.

Journal Requirements:

"Bm received the Medical Humanities Education Grant, 2020 from the University of Toronto. The funders had no role in study design, data collection and analysis, decision to publish, or preparation of the manuscript. "

We note that one or more of the authors is affiliated with the funding organization, indicating the funder may have had some role in the design, data collection, analysis or preparation of your manuscript for publication; in other words, the funder played an indirect role through the participation of the co-authors. If the funding organization did not play a role in the study design, data collection and analysis, decision to publish, or preparation of the manuscript and only provided financial support in the form of authors' salaries and/or research materials, please do the following:

a. Review your statements relating to the author contributions, and ensure you have specifically and accurately indicated the role(s) that these authors had in your study. These amendments should be made in the online form.

b. Confirm in your cover letter that you agree with the following statement, and we will change the online submission form on your behalf: 

“The funder provided support in the form of salaries for authors [insert relevant initials], but did not have any additional role in the study design, data collection and analysis, decision to publish, or preparation of the manuscript. The specific roles of these authors are articulated in the ‘author contributions’ section.

Reviewers' comments:

Reviewer's Responses to Questions

**Comments to the Author**

1. Is the manuscript technically sound, and do the data support the conclusions?

Reviewer #1: Partly

Reviewer #2: Yes

2. Has the statistical analysis been performed appropriately and rigorously? 

Reviewer #1: N/A

Reviewer #2: N/A

3. Have the authors made all data underlying the findings in their manuscript fully available?

Reviewer #1: Yes

Reviewer #2: Yes

4. Is the manuscript presented in an intelligible fashion and written in standard English?

Reviewer #1: Yes

Reviewer #2: Yes

5. Review Comments to the Author

Reviewer #1: The paper consists in an analysis of a selection of narratives published on a single journal.

I found the paper interesting and pleasant to be read; several statements and quotations resonated with my own ED past practice.

However, while the paper seems flow and reasonable, it is lacking on the methodological side, expecially regarding data sources selection. Moreover, current journal policies mandate that Qualitative research must adhere to appropriate study design and reporting guidelines.

Here are some points for the Authors to be considered;

1) No effort has been reported for bias reduction strategies (expecially given the high risk of involuntary or inconscious cherry-picking) in choosing narratives to be included or excluded;

2) Consistently, reported narratives were focused on single episodes ("transformative moments"), usually with recurrent elements of tension (as seen in Appendix 2). However, no explanation is given on the absence of tension-less narratives - do such situations simply not happen, or is there any publication bias towards specific kinds of narrative? Altought this last question does not directly relate in the study objectives, the possibility should be at least mentioned in the dicussion section.

3) No information has been given on the number of narrative excluded from the study, nor on inclusion criteria.

4) Study design is not well defined, lacking a well formalized objective. Analyzing a selection of published clinicians' pieces "to determine which experiences most profoundly impacted their professional identity formation" is a rather ambitious objective given the design did not follow a systematic approach.

5) No quality assesment of retrieved pieces was performed.

6) Authors mentioned to have performed a search on 30 different journals. No methodology for such selection has been described.

7) No effort has been described in expanding selecion in languages different than English, or grey literature; sources such as blogs, articles and papers can be found aswell. For instance, an example of narrative collection in the same field of emergency medicine can be found here after a quick google search: https://litfl.com/anthology/literary-medicine-musings/.

9) No specifing reporting guideline has been used, in contrast with Journal policies. Please adhere to either COREQ, SRQR, or any other applicable EQUATOR guideline.

I believe that none of the mentioned points (beside last one) is per se obstative to article acceptance, however they all contribute to expose the work to a significant risk of bias in its conduction and conclusions.

Reviewer #2: dear, I read with interest your paper. I appreciate your work. many thanks for your work and fot time that you spent. In my personal opinion narrative medicine in not properly considered by clinicians.

6. PLOS authors have the option to publish the peer review history of their article (what does this mean?). If published, this will include your full peer review and any attached files.

Reviewer #1: **Yes: **Antonio Vinci

Reviewer #2: **Yes: **francesco rosiello

---

## [Author Response · Author response to Decision Letter 0]

29 Aug 2021

EDITOR COMMENTS: 

Interesting study that explores a field of Narrative Medicine in Critical Care with a new approach. However, to be publishable on PLOSONE it is necessary that the Authors answer the following points:

1. To respond accurately to the Reviewers’ requests

Response 1: Thank you. We have done that in detail below. 

2. To improve the methodology of the proposed study

Response 2: Our responses to Reviewer 1, our edits, and our inclusion of the “Reporting Guidelines” demonstrate a study with a sound methodology. 

3. To better define the study design

Response 3: We conducted a qualitative analysis, using a narrative methodology and thematic data analysis, we sought primarily to understand how the reflective pieces helped the writer make sense of their key experiences and influenced their professional identity formation. 

4. To highlight how the results of the study can be valid in contexts other than the one where the study was carried out.

Response 4: Using published narratives, our findings were based on analyzing reflections of more than 100 international clinicians across all continents, written through 8 years. We didn’t collect reflections of clinicians in a single institution at a single time point. Our results are therefore valid in many different contexts (spanning over time, space, people). 

5. Please use specifing reporting guideline and adhere to either COREQ, SRQR, or any other applicable EQUATOR guideline.

Response 5: We have included the Table at the very end of this document. 

6. Please Explain in the Discussion the acronym PIF.

Response 6: We have explained Professional Identity Formation (PIF) at the beginning of the Discussion.

7. Please include the Limitations and Conclusions paragraphs in the paper

Response 7: There is a Limitations and Conclusions paragraph in our manuscript. 

8. To improve the bibliography by adding more recent studies on the subject.

Response 8: The following References have been added throughout the manuscript:

Jowsey T, et al. Performativity, identity formation and professionalism: Ethnographic research to explore student experiences of clinical simulation training. PLoS One. 2020 Jul 30;15(7):e0236085.

Wald HS, Ruddy M. Surreal Becomes Real: Ethical Dilemmas Related to the COVID-19 Pandemic and Professional Identity Formation of Health Professionals. J Contin Educ Health Prof. 2021 Apr 1;41(2):124-129.

Stetson GV, Kryzhanovskaya IV, Lomen-Hoerth C, Hauer KE. Professional identity formation in disorienting times. Med Educ. 2020 Aug;54(8):765-766. 

Findyartini A, Anggraeni D, Husin JM, Greviana N. Exploring medical students' professional identity formation through written reflections during the COVID-19 pandemic. J Public Health Res. 2020 Nov 17;9(Suppl 1):1918. 

Khoo SM, Serene WXL. When Faculty Tell Tales: How Faculty Members' Reflective Narratives Impact Residents' Professional Identity Formation. Acad Med. 2021 Jul 27. 

Lawrence EC, Carvour ML, Camarata C, Andarsio E, Rabow MW. Requiring the Healer's Art Curriculum to Promote Professional Identity Formation Among Medical Students. J Med Humanit. 2020 Dec;41(4):531-541. 

REVIEWER #1: 

The paper consists in an analysis of a selection of narratives published on a single journal. I found the paper interesting and pleasant to be read; several statements and quotations resonated with my own ED past practice.

However, while the paper seems flow and reasonable, it is lacking on the methodological side, especially regarding data sources selection. Moreover, current journal policies mandate that Qualitative research must adhere to appropriate study design and reporting guidelines.

Here are some points for the Authors to be considered;

Comment 1: No effort has been reported for bias reduction strategies (especially given the high risk of involuntary or unconscious cherry-picking) in choosing narratives to be included or excluded.

Response 1: 

Thank you. As explained in our methods section in the manuscript we report carefully how we have chosen and included narratives from the only journal consistently publishing narrative pieces since 2013. Furthermore, we did not pick among the 84 pieces from the journal but included them all: “In order to identify pieces published by ICU clinicians, we looked across 30 journals that publish content related to Intensive Care. We looked at each journal’s “Instructions for authors” guidelines to find out if the journal published narrative pieces written by clinicians (Supplemental Digital Appendix 1). We found one journal, that consistently published narrative pieces since 2013 in a section titled “From the Inside”25. These 84 reflective pieces constituted our data source.”

Concerning the comment on our lack of “reporting bias reduction strategy”. Our study follows the ontology of qualitative research and thus a social constructivist view in which personal “bias” is a condition of possibility and the assessment of the quality of research must instead hinge on the transparency as to how data was selected (see for instance Stige et al 2009).

We have described clearly that we were very inclusive of narratives published in Intensive Care journals. We included all the ones that were published therefore we did not “cherry pick”. We have edited the last sentence to more clearly portray that:

We found one journal, that consistently published narrative pieces since 2013 in a section titled “From the Inside”25. We included all 84 reflective pieces published and these pieces constituted our data source.”

Comment 2: Consistently, reported narratives were focused on single episodes ("transformative moments"), usually with recurrent elements of tension (as seen in Appendix 2). However, no explanation is given on the absence of tension-less narratives - do such situations simply not happen, or is there any publication bias towards specific kinds of narrative? Although this last question does not directly relate in the study objectives, the possibility should be at least mentioned in the discussion section.

Response 2:

Thank you for this interesting comment. Narratives are reflective pieces. Reflection is “a term for those intellectual and affective activities in which individuals engage to explore their experiences in order to [gain] a new understanding and appreciation.”, as such narratives are bound to be about transformative moments. We see this comment as related to “comment 1” about bias in selection strategies. We agree that we did make analytical choices in what themes we present as central across the narratives. However, the analytical process of how we found those themes and the large transdisciplinary group (5 of us with different backgrounds) that helped validate the selection of themes is described clearly on the Methods section: “Three research team members: a clinician and educator (BM), a medical anthropologist (LN) and an art historian (KM) participated in the initial coding and data analysis… Once the research team agreed upon the thematic structure a clinician and ethicist (AH) and an ICU trainee (CA) read the pieces for analytic validation, to see if they recognized the same themes as central to the writers’ experiences.” In particular, the choice to focus on tensions as opposed to tension-less narratives followed the tendency in the empirical material in which the majority of the narratives presented narratives centering on tensions around life-and-death (see main theme on the result section).

Additionally, to strengthen the analysis, we presented our findings to our multidisciplinary group (Intensive Care doctors, trainees, nurses and respiratory therapist) to see of the findings resonated with them. In our presentation there were 40 participants and even though they didn’t write these pieces, our findings resonated with all. To strengthen our analysis, we have added this sentence to the result section to include the result of the presentation:

“Member check in was not realistic, considering the pieces were written by more than 100 international clinicians. Our team presented our finding to a group of more than 40 Intensive Care clinicians (doctors, trainees, nurses and respiratory therapists)in our institution and the conclusion was that the finding resonated with them all.” 

Comment 3: No information has been given on the number of narratives excluded from the study, nor on inclusion criteria.

Response 3:

We have clarified in the methods section how we wanted to include all narratives written by Intensive Care Clinicians. We have explained in the response to you in Comment 1 and by editing one of the sentences that we didn’t exclude any narratives (we included all the published pieces). 

Comment 4: Study design is not well defined, lacking a well formalized objective. Analyzing a selection of published clinicians' pieces "to determine which experiences most profoundly impacted their professional identity formation" is a rather ambitious objective given the design did not follow a systematic approach.

Response 4:

Thank you for this comment. We agree that the word “profoundly” is maybe not the right word here, so we changed it to: 

“What are the experiences shared through narratives that impact ICU clinicians’ identity formation?”

We did follow a systematic approach in selection of the pieces as well as our analysis of the pieces. We have clearly described that in our “Material and Methods” and furthermore in our completed table that goes over the checklist of Standards for reporting Qualitative Research. 

Comment 5: No quality assessment of retrieved pieces was performed.

Response 5:

Thank you for this comment. We understand that we have embarked on a quest that has not yet been embarked on. This study is not a classic, quantitative “literature review” and the narratives of these clinicians do not report on original research data and have not applied any research methodology. The only assessment of these pieces is that: they describe key professional moments that have moved the authors and they have gone peer review by the journal that published them. We should be careful not to make “quality assessments” of such personal narratives. Quantifiable markers of achievement could be detrimental to what narratives aims to achieve and as mentioned by some authors “the very essence and purpose of reflection may be compromised when it is enforced in an overly prescriptive manner, or when it is subjected to formal evaluation” (Ng et al. Med Edu 2015; 49: 461-75).

Comment 6: Authors mentioned to have performed a search on 30 different journals. No methodology for such selection has been described.

Response 6:

In our “Materials and Methods” we describe: “In order to identify pieces published by ICU clinicians, we looked across 30 journals that publish content related to Intensive Care. We looked at each journal’s “Instructions for authors” guidelines to find out if the journal published narrative pieces written by clinicians (Supplemental Digital Appendix 1).”

Again, our study is not a conventional Literature Review. Our study is a Qualitative Research Study using qualitative data. We had two choices. We could have asked clinicians in our center to write narratives and personal reflections but that would have been a limited data source and most would not embrace these type of writing. 

Published narratives were then rich sources of data for the following reasons: (a) They are written through almost a decade, (b) they were written from more than 100 clinicians across all continents, (c) the pieces have undergone peer review. 

When deciding to use published narratives we decided to look at medical journals and since we were focused on Critical Care, we looked at journals that published Critical Care content. Other narratives would be books and books being massive texts written by only one person, wouldn’t be rich sources of data, they would represent the view of one person only. Blogs are generally not peer reviewed. 

Our selection of pieces is not a conventional methodology, but it is a very innovative way of getting rich qualitative data that spans geographical boundaries through almost a decade and have been reviewed by peers. 

We have also added the following paragraph in Materials and Methods:

“Narratives are published in medical journals, books, blogs, and different websites. Books are written by one author, and we were interested in the community of critical care clinicians. We didn’t have a good method of identifying blogs and other websites and generally they are not peer reviewed, so we focused in narratives published in medical journals.”

Comment 7: No effort has been described in expanding selecion in languages different than English, or grey literature; sources such as blogs, articles and papers can be found aswell. For instance, an example of narrative collection in the same field of emergency medicine can be found here after a quick google search: https://litfl.com/anthology/literary-medicine-musings/.

Response 8: 

Again, this is not a systematic literature review. In Qualitative study an important issue is whether you have data saturation or not and most certainly we did. If we didn’t have a thematic saturation, we were going to look into other sources of data. In general, most qualitative studies report a thematic saturation with 10-15 interviews, in our case we had 84 reflections written by more than 100 participants. 

Comment 9: No specifying reporting guideline has been used, in contrast with Journal policies. Please adhere to either COREQ, SRQR, or any other applicable EQUATOR guideline.

Response 9:

Thank you! Please see our Table at the end of this document. 

Comment 10: I believe that none of the mentioned points (beside last one) is per se obstative to article acceptance, however they all contribute to expose the work to a significant risk of bias in its conduction and conclusions.

Response 10: 

Thank you! We hope that our responses and edits have clarified your points and have made the article better. 

REVIEWER #2: 

Dear, I read with interest your paper. I appreciate your work. many thanks for your work and fot time that you spent. In my personal opinion narrative medicine in not properly considered by clinicians.

Response:

Thank you! We agree with you fully on the value of Narrative Medicine. 

 Standards for Reporting Qualitative Research (SRQR)* 

http://www.equator-network.org/reporting-guidelines/srqr/

 Page/line no(s).

Title and abstract 

 Title - Concise description of the nature and topic of the study Identifying the study as qualitative or indicating the approach (e.g., ethnography, grounded theory) or data collection methods (e.g., interview, focus group) is recommended 1

 Abstract - Summary of key elements of the study using the abstract format of the intended publication; typically includes background, purpose, methods, results, and conclusions 2-3

Introduction 

 Problem formulation - Description and significance of the problem/phenomenon studied; review of relevant theory and empirical work; problem statement 5-6

 Purpose or research question - Purpose of the study and specific objectives or questions 5-6

Methods 

 Qualitative approach and research paradigm - Qualitative approach (e.g., ethnography, grounded theory, case study, phenomenology, narrative research) and guiding theory if appropriate; identifying the research paradigm (e.g., postpositivist, constructivist/ interpretivist) is also recommended; rationale** 6-7

 Researcher characteristics and reflexivity - Researchers’ characteristics that may influence the research, including personal attributes, qualifications/experience, relationship with participants, assumptions, and/or presuppositions; potential or actual interaction between researchers’ characteristics and the research questions, approach, methods, results, and/or transferability 7

 Context - Setting/site and salient contextual factors; rationale** n/a

 Sampling strategy - How and why research participants, documents, or events were selected; criteria for deciding when no further sampling was necessary (e.g., sampling saturation); rationale** 6

 Ethical issues pertaining to human subjects - Documentation of approval by an appropriate ethics review board and participant consent, or explanation for lack thereof; other confidentiality and data security issues n/a

 Data collection methods - Types of data collected; details of data collection procedures including (as appropriate) start and stop dates of data collection and analysis, iterative process, triangulation of sources/methods, and modification of procedures in response to evolving study findings; rationale** 6-7

 Data collection instruments and technologies - Description of instruments (e.g., interview guides, questionnaires) and devices (e.g., audio recorders) used for data collection; if/how the instrument(s) changed over the course of the study n/a

 Units of study - Number and relevant characteristics of participants, documents, or events included in the study; level of participation (could be reported in results) 6

 Data processing - Methods for processing data prior to and during analysis, including transcription, data entry, data management and security, verification of data integrity, data coding, and anonymization/de-identification of excerpts 6-7

 Data analysis - Process by which inferences, themes, etc., were identified and developed, including the researchers involved in data analysis; usually references a specific paradigm or approach; rationale** 6-7

 Techniques to enhance trustworthiness - Techniques to enhance trustworthiness and credibility of data analysis (e.g., member checking, audit trail, triangulation); rationale** 7

Results/findings 

 Synthesis and interpretation - Main findings (e.g., interpretations, inferences, and themes); might include development of a theory or model, or integration with prior research or theory 7-15

 Links to empirical data - Evidence (e.g., quotes, field notes, text excerpts, photographs) to substantiate analytic findings 9/11/13/15

Discussion 

 Integration with prior work, implications, transferability, and contribution(s) to the field - Short summary of main findings; explanation of how findings and conclusions connect to, support, elaborate on, or challenge conclusions of earlier scholarship; discussion of scope of application/generalizability; identification of unique contribution(s) to scholarship in a discipline or field 15-19

 Limitations - Trustworthiness and limitations of findings 18

Other 

 Conflicts of interest - Potential sources of influence or perceived influence on study conduct and conclusions; how these were managed 1

 Funding - Sources of funding and other support; role of funders in data collection, interpretation, and reporting 1

 *The authors created the SRQR by searching the literature to identify guidelines, reporting standards, and critical appraisal criteria for qualitative research; reviewing the reference lists of retrieved sources; and contacting experts to gain feedback. The SRQR aims to improve the transparency of all aspects of qualitative research by providing clear standards for reporting qualitative research. 

 **The rationale should briefly discuss the justification for choosing that theory, approach, method, or technique rather than other options available, the assumptions and limitations implicit in those choices, and how those choices influence study conclusions and transferability. As appropriate, the rationale for several items might be discussed together. 

 Reference: 

 O'Brien BC, Harris IB, Beckman TJ, Reed DA, Cook DA. Standards for reporting qualitative research: a synthesis of recommendations. Academic Medicine, Vol. 89, No. 9 / Sept 2014

DOI: 10.1097/ACM.0000000000000388

---

## [Decision Letter · Decision Letter 1]

2 Nov 2021

Who Am I? Narratives as a Window to Transformative Moments in Critical Care

PONE-D-21-09314R1

Dear Dr. Briseida Mema,

We’re pleased to inform you that your manuscript has been judged scientifically suitable for publication and will be formally accepted for publication once it meets all outstanding technical requirements.

Kind regards,

Filomena Pietrantonio

Academic Editor

PLOS ONE

Additional Editor Comments (optional):

The authors have addressed all comments in a satisfactory way.

Reviewers' comments:

Reviewer's Responses to Questions

**Comments to the Author**

1. If the authors have adequately addressed your comments raised in a previous round of review and you feel that this manuscript is now acceptable for publication, you may indicate that here to bypass the “Comments to the Author” section, enter your conflict of interest statement in the “Confidential to Editor” section, and submit your "Accept" recommendation.

Reviewer #1: All comments have been addressed

Reviewer #2: All comments have been addressed

2. Is the manuscript technically sound, and do the data support the conclusions?

Reviewer #1: Yes

Reviewer #2: Yes

3. Has the statistical analysis been performed appropriately and rigorously? 

Reviewer #1: N/A

Reviewer #2: N/A

4. Have the authors made all data underlying the findings in their manuscript fully available?

Reviewer #1: Yes

Reviewer #2: Yes

5. Is the manuscript presented in an intelligible fashion and written in standard English?

Reviewer #1: Yes

Reviewer #2: Yes

6. Review Comments to the Author

Reviewer #1: The authors have addressed all comments in a satisfactory way

I believe the paper can be accetted for publication.

Reviewer #2: thank you for your work. I appreciate the revised draft. I hope to read the published article soon!!

7. PLOS authors have the option to publish the peer review history of their article (what does this mean?). If published, this will include your full peer review and any attached files.

Reviewer #1: **Yes: **Antonio Vinci

Reviewer #2: No

---

## [Editor Report · Acceptance letter]

4 Nov 2021

PONE-D-21-09314R1 

* Who Am I? * Narratives as a Window to Transformative Moments in Critical Care 

Dear Dr. Mema:

I'm pleased to inform you that your manuscript has been deemed suitable for publication in PLOS ONE. Congratulations! Your manuscript is now with our production department. 

Kind regards, 

on behalf of

Dr. Filomena Pietrantonio 

Academic Editor

PLOS ONE